# Age-Related microRNA Overexpression in Lafora Disease Male Mice Provides Links between Neuroinflammation and Oxidative Stress

**DOI:** 10.3390/ijms24021089

**Published:** 2023-01-06

**Authors:** Carlos Romá-Mateo, Sheila Lorente-Pozo, Lucía Márquez-Thibaut, Mireia Moreno-Estellés, Concepción Garcés, Daymé González, Marcos Lahuerta, Carmen Aguado, José Luis García-Giménez, Pascual Sanz, Federico V. Pallardó

**Affiliations:** 1Department of Physiology, Facultat de Medicina i Odontologia, Universitat de València, 46010 Valencia, Spain; 2Fundación Instituto de Investigación Sanitaria INCLIVA, 46010 Valencia, Spain; 3Centro de Investigación Biomédica en Red de Enfermedades Raras (CIBERER)—ISCIII, 46010 Valencia, Spain; 4Neonatal Research Group, Health Research Institute La Fe, 46026 Valencia, Spain; 5Institut d’Investigació Biomèdica de Girona Dr. Josep Trueta (IDIBGI), Parc Hospitalari Martí i Julià de Salt, 17190 Girona, Spain; 6Instituto de Biomedicina de Valencia, Consejo Superior de Investigaciones Científicas, 46010 Valencia, Spain; 7EpiDisease S.L. (Spin-off From the CIBER-ISCIII), Parc Científic de la Universitat de València, 46980 Paterna, Spain; 8Novartis Institutes for BioMedical Research (NIBR), Novartis Campus, CH-4056 Basel, Switzerland

**Keywords:** epilepsy, microRNA, gene expression, neuroinflammation

## Abstract

Lafora disease is a rare, fatal form of progressive myoclonus epilepsy characterized by continuous neurodegeneration with epileptic seizures, characterized by the intracellular accumulation of aberrant polyglucosan granules called Lafora bodies. Several works have provided numerous evidence of molecular and cellular alterations in neural tissue from experimental mouse models deficient in either laforin or malin, two proteins related to the disease. Oxidative stress, alterations in proteostasis, and deregulation of inflammatory signals are some of the molecular alterations underlying this condition in both KO animal models. Lafora bodies appear early in the animal’s life, but many of the aforementioned molecular aberrant processes and the consequent neurological symptoms ensue only as animals age. Here, using small RNA-seq and quantitative PCR on brain extracts from laforin and malin KO male mice of different ages, we show that two different microRNA species, miR-155 and miR-146a, are overexpressed in an age-dependent manner. We also observed altered expression of putative target genes for each of the microRNAs studied in brain extracts. These results open the path for a detailed dissection of the molecular consequences of laforin and malin deficiency in brain tissue, as well as the potential role of miR-155 and miR-146a as specific biomarkers of disease progression in LD.

## 1. Introduction

Lafora disease (LD) is a rare genetic condition (OMIM254780, ORPHA#501) caused by either mutations in the locus *EPM2A*, coding for the dual specificity phosphatase laforin, or *EPM2B*, coding for the E3 ubiquitin ligase malin [1,2]. LD is a fatal progressive myoclonus epilepsy characterized by tonic–clonic seizures, myoclonus, visual hallucinations, ataxia, and dementia, followed by a rapid neurodegenerative progression that results in the death of patients with a median disease duration of 11 years [3] since the onset of the first symptoms [4,5,6]. There is no cure for this terrible disease beyond palliative treatments; however, over the last few years, research has produced numerous molecular data that provide accumulating evidence of its pathophysiology. For instance, the accumulation of aberrant glycogen intracellular deposits, called Lafora bodies (LBs), has attracted most of the focus in LD research, and constitutes the target of many therapeutical strategies proposed so far. However, molecular impairment of laforin and/or malin, which work as a functional complex (involved in many molecular pathways not only related to glycogen metabolism but also to proteostasis pathways and the ubiquitin–proteasome system) leads to many cellular alterations that seem independent of the progression of LBs accumulation and their consequences for neuronal survival [7,8].

Recent research has highlighted the role of glia in the pathophysiology of the disease, reporting impairment in the regulation of the glutamate transporter in astrocytes from mouse models of LD [9]. We recently provided an exhaustive analysis of the transcriptomic profile in the brains of both *Epm2a* and *Epm2b* knockout (KO) mouse models [10], which showed alterations in the expression of genes related to immune response, depicting a neuroinflammatory landscape that is in agreement with the results from other previous studies, in which inflammatory markers were found to be altered in different brain regions [11,12]. Although most of these gene expression alterations could be explained by the misfunctioning/deficiency in laforin and/or malin, little is known about the relationship between specific regulation of gene expression and the laforin–malin complex. Ganesh and collaborators proposed that malin interacted with RNA processing machinery, thus participating in the regulation of microRNA processing [13], but no other work has offered proof that microRNA-mediated signaling could be altered in LD. In the present work, we provide evidence for the overexpression of two microRNA species, namely, miR-146a and miR-155, which are overexpressed in both LD mouse models. Expression of these two microRNAs, which have previously been related to neuroinflammation and epilepsy [14,15,16,17], increases with age, reaching a maximum expression in 16-month-old (mo) mice. The increase in these microRNA levels correlates with an increase in the gene expression of putative target genes and other related genetic regulators directly linked to oxidative stress and inflammatory responses. This study, to our knowledge, is the first to report evidence that microRNAs are altered in the brains of mouse models of LD, and provides interesting pathophysiological clues about the genetic expression alterations registered in these animal models, suggesting that they could potentially be used as future therapeutical targets.

## 2. Results and Discussion

### 2.1. Differential microRNA Expression in the Brain of 16-Month-Old Epm2a−/− and Epm2b−/− Mice as Compared to Control Mice

Total RNA extracts from brain homogenates of 16-month-old mice, previously used for a transcriptomics study [10], were analyzed in a small RNA-seq procedure. Groups compared included samples from *Epm2a−/−* and *Epm2b−/−* animals, as well as wild type controls (*n* = 4 for all groups). After processing and filtering sequences, the number and quality of reads were considered sufficient for analysis (average reads, 5,785,774; max, 7,998,128; min, 3,694,571), and principal component analysis (PCA) was performed (Appendix A). This analysis showed an overall strong degree of clustering for all groups; however, one sample from each of the groups appeared separated from the rest. Thus, differential microRNA expression analysis was performed, either keeping or removing those outlier samples, but no significant changes were observed. Surprisingly, differential microRNA expression analysis highlighted only two microRNA species that were differentially expressed (with a false discovery rate, FDR, < 0.05) between control and both *Epm2a−/−* and *Epm2b−/−* samples: miR-146a (with a FC of 2.17 and 2.13 in *Epm2a−/−* and *Epm2b−/−* vs. control, respectively) and miR-155 (with an FC of 3.81 and 3.41 in *Epm2a−/−* and *Epm2b−/−* vs. control, respectively), with a downregulation of miR-10b only between *Epm2b−/−* and control samples (FC 0.42) (Table 1). Accordingly, no significant differences were found between *Epm2a−/−* and *Epm2b−/−* samples. Given the strong *p*-values and FDRs of these microRNAs shared by the two mutant genotypes, as compared to miR-10b (see Table 1), this latter microRNA was discarded in further analysis.

Both miR-146a and miR-155 have been previously described as epilepsy-related microRNAs that are functionally related to neuroinflammation [15]; this prompted us to further validate the high expression levels registered in the small RNA-seq analysis by specific amplification using real-time quantitative PCR (qRT-PCR) on brain RNA extracts from LD mice models. Detection of both microRNAs was performed on total RNA samples from the same mice used for small RNA-seq analysis, as well as four more total RNA samples obtained from four independent mice from each genotype. In all cases, LD samples showed significant overexpression of both miR-146a (mean value 2.446 ± 0.56 SD; *p*< 0.001, for *Epm2a−/−* mice and 2.327 ± 0.64; *p* < 0.001 for *Epm2b−/−*) and miR-155 (mean value 3.882 ± 1.09 SD; *p*< 0.001, for *Epm2a−/−* mice and 3.878 ± 0.58; *p* < 0.001 for *Epm2b−/−*) microRNAs, corroborating the small RNA-seq data (Figure 1).

### 2.2. MicroRNAs miR-155 and miR-146a Are Overexpressed in an Age-Dependent Manner

Previous studies have shown that most of the features of the pathological phenotype in LD mice models usually increase with age, paralleling disease progression in humans; our previous work proved that the neuroinflammatory landscape in LD mice models becomes evident, especially around 12 months of age [10]. Based on these data, we sought o determine whether the observed increase in the expression levels for both microRNAs was also a late-onset event, related to this progressive inflammatory disbalance, or, on the other hand, a more preliminary one, closer to the onset of the first symptoms. With this aim, we compared the expression levels in brain extracts from mice at 3, 7, 12, and 16 months of age. As shown in Figure 2, the expression levels start to become significantly higher in 12-month-old *Epm2b−/−* mice, and reach a maximum at 16 months for both *Epm2a−/−* and *Epm2b−/−* mice (see Appendix A for the complete set of data).

These data are in agreement with our previous results on transcriptomic data from the same animal models, which showed neuroinflammation markers increasing as animals grew older [10]. Provided the extensive analysis of gene expression therein performed, we searched for putative target genes for miR-146a and miR-155 that showed reduced expression levels in our previous analysis.

### 2.3. Analysis of the Expression of Putative Gene Targets of miR-146a and miR-155 in Brain Extracts from Epm2a−/− and Epm2b−/− Mice of 16 Months of Age

We searched for miR-146a and miR-155 targets in miRTarbase (http://mirtarbase.mbc.nctu.edu.tw/); in parallel, we searched for miRNA–target interactions using the TargetScan (http://www.targetscan.org/ (accessed on 30 May 2018)) prediction algorithm, and for targets of the corresponding microRNAs according to an extensive search of the existing literature. From the combined results of these complementary searches, we selected a set of putative target genes to be further validated, as follows: Nsun3 and Socs1 for miR-155; Glra2, Btg2, Traf6, and Sod2 for miR-146a. Surprisingly, not only did we not obtain a significant decrease in the expression levels of these genes (except for Btg2, which was slightly downregulated in *Epm2a−/−* mice as compared to WT animals), but we registered an increase in the gene expression levels of Nsun3 and Socs1 in both KO mice models (Figure 3a), an increase in Glra2 in both KO mice models, and, finally, an increase in Sod2 and Traf6 only for the *Epm2b−/−* mice (Figure 3b). Regarding Btg2, only a slight but significant decrease in the expression levels was registered in the *Epm2a−/−* mice. All the genes analyzed are somehow related to molecular pathways relevant to the pathophysiological context of LD: Nsun3 (NOP2/Sun RNA methyltransferase 3) encodes a mitochondrial methyltransferase related to encephalomyopathy and seizures [18]; Glra2 (glycine receptor alpha 2) encodes for a glycine receptor that has been predicted to influence cognition, learning, and memory in the developing brain [19]; Btg2 (BTG anti-proliferation factor 2) is an anti-proliferative gene that, interestingly, has been postulated as a neuroprotective gene in mouse models of Huntington’s disease [20]; and Traf6 (TNF receptor-associated factor 6) is an immune response regulator with an E3 ubiquitin ligase activity that has been shown to interact with Sod1 in rat models of amyotrophic lateral sclerosis (ALS), promoting aggregation [21].

Noteworthy is the marked overexpression observed for the mitochondrial dismutase Sod2 (superoxide dismutase 2) and Socs1 (suppressor of cytokine signaling 1), since they are genes related to oxidative stress and inflammation, two hallmarks of LD [8,11], and they have been previously linked to the molecular pathways in which these microRNAs are involved [15,22]. Although the interaction between microRNA and their targets often leads to the downregulation of the latter, there is evidence of a correlation between high microRNA levels together with an overexpression of their targets [23,24]. Since Sod2 and Socs1 are response genes for oxidative stress and inflammatory cascades, respectively, it is possible that the continued stressed state of 16 mo KO mice could lead to an overexpression of the genes herein analyzed, that parallels the overexpression of miR-155 as part of the proinflammatory response [22] and the concomitant overexpression of miR-146a, which has been described to exert an anti-inflammatory role in several pathologies that involve immunity responses [25,26,27]. Socs1 is a negative regulator of cytokine signaling, being induced by the latter, thus pointing to a role in counteracting the effect of miR-155 and the cytokine cascade induced with age in LD animal models (as described in [10]). It should be noted that miR-155 has been proposed as an activator of autophagy, and in hepatic cells infected with hepatitis B virus, miR-155 transfection activated autophagy via the impairment of Socs1/Akt/mTOR axis [28]; miR-155 has also been reported to regulate autophagy and lysosome function in alcoholic liver disease [29], as well as the inhibition of mitophagy [30]. Since, in LD, there is evidence of impaired autophagy [31,32,33] and mitophagy [34], our results showing both an overexpression of miR-155 and Socs1 could suggest an attempt to promote LB degradation in LD mice. These data suggest a potential therapeutic role for miR-155 and miR-146a for exogenous manipulation, since both microRNAs have been previously related to neuroinflammation in epileptogenic animal models [15,35], and they have even been proposed as potential therapeutic targets for epilepsy [15,36,37]. It should be noted, however, that the results herein reported have been entirely obtained from male mice; further investigation should be carried out to conclude whether the alterations observed in these animal models are generalized or sex-dependent. To our knowledge, nonetheless, this is the first case in which both microRNAs are described as altered in LD animal models, together with an altered gene expression of putative gene targets related to oxidative stress, autophagy, and inflammation; thus, these microRNAs warrant potential for further investigation, which may lead to the definition of their specific role as biomarkers for LD progression or even as potential therapeutic targets.

## 3. Materials and Methods

### 3.1. Animal Care, Mice, and Husbandry

Homozygous male *Epm2a−/−* and *Epm2b−/−* mice in a pure C57BL/6JRccHsd background and the corresponding controls of 3, 7, 12, and 16 months of age were used in this study. The presence of polyglucosan inclusions (the hallmark of LD) in the brain samples of the *Epm2a−/−* and *Epm2b−/−* mouse lines was confirmed by PAS (periodic acid–Schiff) staining. Mice were maintained in the IBV-CSIC facility on a 12/12 light/dark cycle under constant temperature (23 °C) with food and water provided ad libitum. Only male animals were used in this work in order to compare the small RNA-seq results with previous results obtained by RNA-seq analysis [10]. Male mice were sacrificed by cervical dislocation, and whole brains rinsed twice with cold PBS (phosphate-buffered saline) and frozen immediately in liquid nitrogen. When planning the experiments, the principles outlined in the ARRIVE guidelines and the Basel declaration, including the 3R concept, were considered.

### 3.2. Whole RNA Extraction from Mouse Brain

Frozen brains were pulverized with a liquid-nitrogen-cooled stainless steel mortar and pestle. Then, 1 mL of TRIzol™ Reagent (Thermo Fisher Scientific, Madrid, Spain) was added per 100 mg of pulverized brain and homogenized by pipetting up/down five times with a 1 mL tip to break tissue into small pieces. The homogenates were passed ten times through a 21 gauge needle in 2 mL syringe and incubated for 5 min at room temperature (RT) on a plate shaker to allow complete lysis. Samples were centrifuged for 10 min at 12,000× *g* and 4 °C. Supernatants were transferred to a new tube. Next, 0.2 mL of chloroform per 1 mL of TRIzol™ Reagent was added to each tube, mixed by inversion ten times, and incubated for 3 min at room temperature (RT). Samples were centrifuged at 12,000× *g* for 15 min at 4 °C, and the top aqueous phase (0.6 mL) was transferred to a new tube. Finally, RNAs were precipitated by adding 1 volume of isopropanol to the aqueous phase, mixed by inversion, and incubated for 10 min at RT. Samples were centrifuged for 15 min at 12,000× *g* and 4 °C. RNA pellets were washed twice with ethanol 75%, air dried, and resuspended in 75 µL RNase-free water. RNA purity was assessed using a NanoDropONE spectrophotometer (Thermo Fisher, Madrid, Spain), and quality was evaluated using an RNA 6000 Nano Kit and Agilent 2100 Bioanalyzer System (Agilent, Madrid, Spain).

### 3.3. Small RNA-Seq and Data Analysis

The small RNA-seq experiment was conducted in the Multigenic Analysis Unit of the UCIM-INCLIVA (University of Valencia, Valencia Spain). First, 1 µg of total RNA was used for library preparation. cDNA libraries were prepared using the TruSeq Stranded mRNA LP (48 Spl) (Illumina, San Diego, CA), according to the manufacturer’s instructions. The concentration of the cDNA libraries was measured using Qubit 3.0 and the Qubit dsDNA HS Assay Kit (Thermo Fisher Scientific, Waltham, MA, USA). The length distribution of library fragments was determined using the Agilent 2100 Bioanalyzer and the Agilent DNA 1000 Kit (Agilent, Santa Clara County, CA, USA). Libraries were then sequenced on the Illumina^®^ NextSeq 550 to generate 75 bp single reads using the Illumina^®^ NSQ 500 High Output KT v2 (75 CYS) (Illumina, San Diego, CA, USA), according to the manufacturer’s protocol.

Data analysis was performed with the support of EpiDisease S.L. (spin-off from the Center for Biomedical Network Research, CIBER-SCIII, Spain). All reads were aligned using the mouse reference genome (GRCm38 version) from Ensembl. Subsequently, the number of reads corresponding to mature mouse microRNAs was obtained using miRbase v21. The mapping and quantification steps were performed using the Subread and Rsubread packages (http://subread.sourceforge.net/), which comprise a suite of high-performance software programs for processing next-generation sequencing data. microRNAs with very low counts across all libraries provide little evidence for differential expression. These microRNAs were filtered out prior to further analysis: each microRNA needed to have more than one count per million (CPM) in at least four samples (the size of the smallest group) to be considered, otherwise, the expression of the gene was discarded. Subsequently, the trimmed mean of M-values (TMM) normalization [34] was performed to eliminate composition biases between libraries. The specific dispersions per microRNA were estimated with the weighted empirical Bayes probability method [35]. The differential expression analysis was executed using a quasi-likelihood F-test [36]. Raw *p*-values were corrected for multiple testing using the Benjamini–Hochberg method and the FDR (false discovery rate) calculated accordingly [37]. Before carrying out the differential expression analysis, data were explored by generating a multidimensional scaling (MDS) plot. This visualizes the differences between the expression profiles of different samples in two dimensions. The number of unique and common differentially expressed microRNAs in Epm2a−/− and Epm2b−/− samples compared to the control group was represented in Venn diagrams. In addition, samples were hierarchically clustered by their gene expression pattern similarity and represented in a heatmap. Volcano plots were used to represent the proportion of differentially expressed microRNAs obtained from the Epm2a−/− vs. control and Epm2b−/− vs. control comparisons. The 30 most upregulated microRNAs in both comparisons (FC > 2 and FDR < 0.01) were labeled.

### 3.4. RT-qPCR Analyses

Complementary DNA (cDNA) was prepared from total RNA extracts obtained as described above, using the TaqMan™ microRNA Reverse Transcription Kit (Applied Biosystems, Life Technologies, Foster City, CA, USA) and the specific oligonucleotides from the Taqman™ MicroRNA Assay Kit (Applied Biosystems) for endogenous controls RNU6 and sno234, and the target microRNAs, miR-146a and miR-155, following the manufacturer’s recommendations, adapting the calculations to introduce 50 ng/μL of RNA in a 20 μL reaction volume. For each 20 μL of master mix, 2 μL of RNA and 3.35 μL of nuclease-free water were mixed with 8 μL of a pool of specific Taqman™ MicroRNA RT Primers (5x), 0.4 μL of dNTPs (100 mM), 4 μL of MultiScribe reverse transcriptase (50 U/μL), 2 μL of 10X reverse transcription buffer, and 0.25 μL of RNase inhibitor (20 U/μL) (Applied Biosystems). The parameters used to carry out the reactions were those indicated by the manufacturer. A negative control with nuclease-free water was included to ensure no contamination with genomic DNA. For the real-time quantitative PCR, three technical replicates of each cDNA sample were generated for each microRNA to be tested, all in a 384-well plate using a QuantStudio 5 Real-Time PCR System thermocycler (Thermo Fisher Scientific). For each 10 μL of master mix, 1 μL of cDNA and 3.5 μL of nuclease-free water were mixed with 0.5 μL Taqman Small RNA Assay (20x) (specific for each microRNA) and 5 μL of 2× TaqMan Universal Master Mix II (Applied Biosystems). The TaqMan™ Universal Master Mix II non-UNG Kit (Applied Biosystems) and the oligonucleotides corresponding to the controls and microRNA of the previous reaction (Applied Biosystems) were used. The reactions were carried out using the following parameters: 95 °C for 10 min, 40 cycles of 95 °C for 15 s, and 60 °C for 1 min.

For gene expression analysis of putative target genes, cDNA was prepared from total RNA extracts using the High-Capacity cDNA Reverse Transcription Kit (Applied Biosystems, Life Technologies, Foster City, CA, USA) following the manufacturer’s recommendations. We adapted the calculations for introducing 200 ng/μL of RNA in a 20 μL reaction volume. For a single sample, the master mix preparation comprised 10 μL of RNA, 2 μL of pool RT primers (5x), 0.8 μL of dNTPs (100 mM), 1 μL of MultiScribe reverse transcriptase (50 U/μL), 2 μL of 10X reverse transcription buffer, and 4.2 μL of nuclease-free water. A negative control of nuclease-free water was included to ensure no contamination with genomic DNA. The parameters for thermal cycling were 10 min at 25 °C, 120 min at 37 °C, and 5 min at 85 °C. For the real-time quantitative PCR, three technical replicates of each cDNA sample were generated for each gene to be tested, all in a 384-well plate using a QuantStudio 5 Real-Time PCR System thermocycler (Thermo Fisher Scientific). For each 10 μL of master mix, 1 μL of cDNA and 3.5 μL of nuclease-free water was mixed with 0.5 μL TaqMan^®^ Gene Expression Assay (20x) (specific for each gene: Nsun3 (Mm01169906_m1), Socs1 (Mm00782550_s1), Glra2(Mm01168376_m1), Btg2 (Mm00476162_m1) Traf6 (Mm00493836_m1), Sod2 (Mm01313000_m1), and Gapdh (Mm99999915_g1)), and 5 μL of TaqMan Gene Expression Master Mix (Applied Biosystems). The reactions were carried out using the following parameters: 50 °C for 2 min, 95 °C for 10 min, 40 cycles of 95 °C for 15 s, and 60 °C for 1 min.

Finally, all RT-qPCR data were captured using the QuantStudio™ Design and Analysis Software (version 1.5.1, Thermo Fisher). The relative expression of each sample was calculated using the 2^−ΔΔCT^ method (Livak & Schmittgen, 2001), using the CT (cycle threshold) value of RNU6 or sno234 in microRNA analysis, or Gapdh in gene expression analysis, as an internal control for the normalization of the CT of each sample. In the case of microRNA analysis, similar values were obtained for both endogenous controls, with no significant differences in their average CT values. Graphical representations and statistical analyses were conducted using GraphPad Prism v.8.3.0 for Windows (GraphPad Software, San Diego, CA, USA, www.graphpad.com).

## Figures and Tables

**Figure 1 ijms-24-01089-f001:**
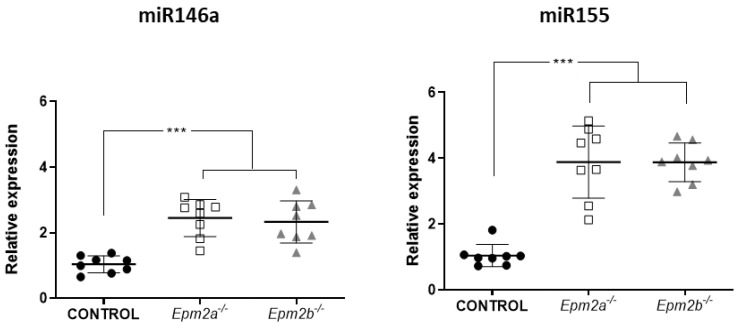
miR-146a and miR-155 are overexpressed in brain extracts from 16-month-old *Epm2a−/−* and *Epm2b−/−* mice. Total RNA was extracted from brain homogenates of 16-month-old WT (CONTROL) and LD KO mice (*Epm2a*−/−, *Epm2b−/−*), and quantitative real-time PCR analyses were performed using the TaqMan™ probe system. The relative expression of both microRNAs was determined using the comparative 2^−ΔΔCT^ method with the RNU6 gene as an internal reference. *** *p* < 0.001 when compared to control mice via two-way ANOVA (*n* = 8).

**Figure 2 ijms-24-01089-f002:**
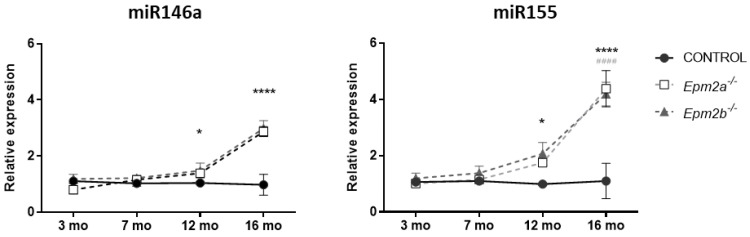
Expression of miR-146a and miR-155 increases with age. Total RNA was extracted from brain homogenates of 3 mo, 7 mo, 12 mo, and 16 mo WT and LD KO mice (*Epm2a−/−, Epm2b−/−*), and quantitative real-time PCR analyses were performed using the TaqMan™ probe system. The relative expression of both microRNAs was determined using the comparative 2^−ΔΔCT^ method with the RNU6 gene as an internal reference. Statistical significance was calculated as #### *p* < 0.0001 for *Epm2a −/−* mice, and * *p* < 0.05, **** *p* < 0.0001 for *Epm2b−/−* mice, when, in both cases, compared to control mice via two-way ANOVA (*n* = 8).

**Figure 3 ijms-24-01089-f003:**
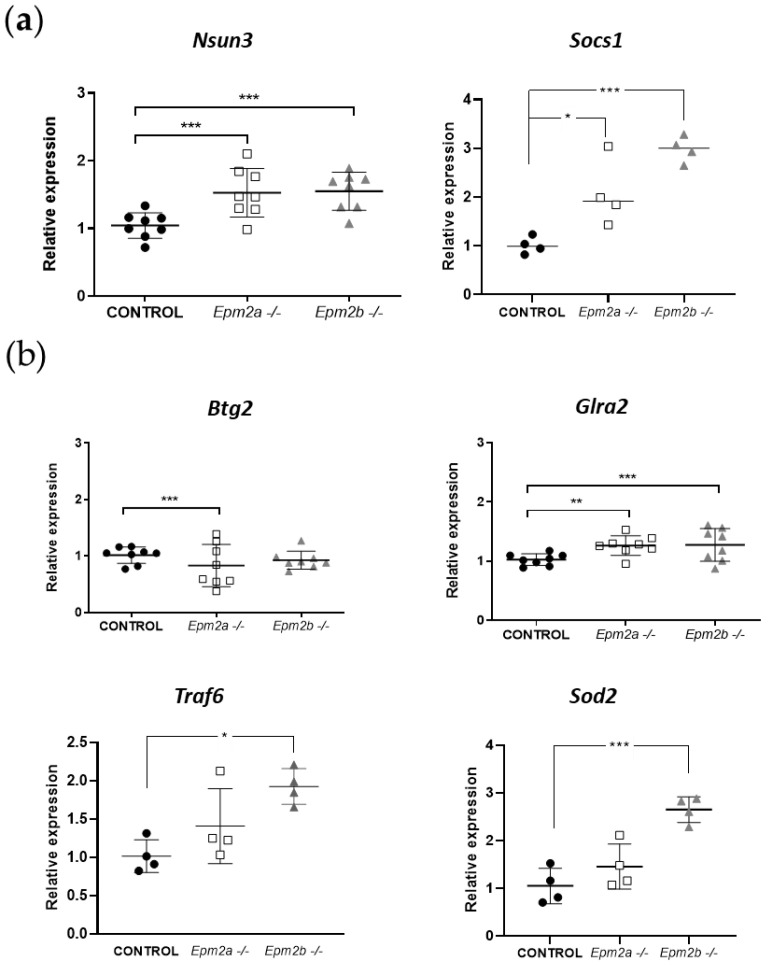
Expression of miR-146a/miR-155 putative target genes for miR-155 (**a**) and for miR-146a (**b**) in the brains of 16 mo mice. Total RNA was extracted from brain homogenates of 16 mo WT (CONTROL) and LD KO mice (*Epm2a−/−, Epm2b−/−*), and quantitative real-time PCR analyses were performed using the TaqMan™ probe system. The relative expression of putative target genes was determined using the comparative 2^−ΔΔCT^ method with Gapdh gene as an internal reference. * *p* < 0.05, ** *p* < 0.01, *** *p* < 0.001, when compared to control mice via two-way ANOVA (*n* = 4 or *n* = 8 when indicated).

**Table 1 ijms-24-01089-t001:** Differentially expressed microRNAs after small RNA-seq analysis.

** *Epm2a^−/−^* ** ** *vs. Control* **	**Log2FC/FC**	** *p* ** **-value**	**FDR**
*mmu-miR-155-5p*	3.81	1.366 × 10^−7^	8.933 × 10^−5^
*mmu-miR-146a-5p*	2.17	8.018 × 10^−7^	2.622 × 10^−4^
** *Epm2b^−/−^* ** ** *vs. control* **	**Log2FC/FC**	** *p* ** **-value**	**FDR**
*mmu-miR-155-5p*	3.41	5.194 × 10^−7^	3.397 × 10^−4^
*mmu-miR-146a-5p*	2.13	1.138 × 10^−6^	3.720 × 10^−4^
*mmu-miR-10b-5p*	0.42	1.933 × 10^−4^	4.213 × 10^−2^

## Data Availability

Not applicable.

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
