# Peer review of "Age-Related microRNA Overexpression in Lafora Disease Male Mice Provides Links between Neuroinflammation and Oxidative Stress"

_ijms, 2023, doi:10.3390/ijms24021089_

Round 1
Reviewer 1 Report
Main point
Despite recognition of the importance of sex and gender in most areas of research, important knowledge gaps persist owing to the general orientation of scientific attention to one sex category. The gap in the representation of females in studies on human subjects, but also in animal models has been well documented. The title should indicate that the study is carried out on an animal model. If only one sex is included in the study, the title and the abstract should specify the sex of animals. Authors should indicate why the study model was based on one sex. The article’s title should specify this fact by including “in males” in the title
Moreover, the authors should justify the reasons for any exclusion of female mice. Methodological choices about sex in relation to your study should be reported and justified in the same way as other methodological choices. In discussion the authors should further discuss the implications of the lack of female mice analysis on the interpretation of the results. If the authors retain a limit to have selected only male animals or if the results could be explained also for females
Minor points
Results
Page 3 Line 106: mean±SD and p value of miR146a and miR155 levels should be added in the text.
Page 5 Fig. 3 Why some experiments were only carried out in four animals, such as for Traf6, Socs1 and Sod 2?
Author Response
Despite recognition of the importance of sex and gender in most areas of research, important knowledge gaps persist owing to the general orientation of scientific attention to one sex category. The gap in the representation of females in studies on human subjects, but also in animal models has been well documented. The title should indicate that the study is carried out on an animal model. If only one sex is included in the study, the title and the abstract should specify the sex of animals. Authors should indicate why the study model was based on one sex. The article’s title should specify this fact by including “in males” in the title.
Moreover, the authors should justify the reasons for any exclusion of female mice. Methodological choices about sex in relation to your study should be reported and justified in the same way as other methodological choices. In discussion the authors should further discuss the implications of the lack of female mice analysis on the interpretation of the results. If the authors retain a limit to have selected only male animals or if the results could be explained also for females.
We agree with the interesting point raised by the reviewer on the use of animals of both sexes in the experimental approach and the necessity to explain and clarify if otherwise. In this case, we only used male animals because we wanted to compare the results of this work with previous results obtained by RNA-seq analysis (Lahuerta et al., 2020), which was performed only with male animals. We have thus changed the title to “Age-related microRNA overexpression in Lafora disease male mice provides links between neuroinflammation and oxidative stress”, following the suggestion of the reviewer. We have also highlighted the sex of the animals in the abstract: “Here, using smallRNA-seq and quantitative PCR in brain extracts from laforin and malin KO male mice of different ages, we show that two different microRNA species, miR-155…” (lines 27-28).
Regarding the Methodology, we have also modified lines 231: “Homozygous male Epm2a-/- and Epm2b-/- mice in a pure C57BL/6JRccHsd background…” and 236-238: “Only male animals were used in this work in order to compare the small RNA-Seq results with previous results obtained by RNA-seq analysis [10].”.
Finally, we have included the following paragraph as part of the discussion (lines 220-223): “It should be noted, however, that the results herein reported have been entirely performed in male mice; further investigation should be carried out to conclude if the alterations observed in these animal models are generalized or sex dependent.”
We hope that these amendments will make our results clearer and more rigorous avoiding further confusion, and we want to thank again the reviewer for pointing out such a relevant issue.
Regarding the two minor points raised by the reviewer:
Page 3 Line 106: mean±SD and p value of miR146a and miR155 levels should be added in the text.
Following the reviewer’s suggestion, we have added the specific value of these parameters in the main text (see lines 117-119).
Page 5 Fig. 3 Why some experiments were only carried out in four animals, such as for Traf6, Socs1 and Sod 2?
We thank the reviewer for this insightful comment: the reason for this discrepancy in the number of animals used is simply methodological; the first genes analysed were performed using the remnants of RNA samples used for the small-RNA-Seq analysis at the beginning of the project (only 4 animals were subject to Small-RNA Seq analysis); we then added samples from newly-bred animals of the same genetic background to reinforce the results observed; however, when the remnants were finished, we could only repeat the rest of analysis with RNA from the four newly-bred mice, and since we observed that the differences in genes like Socs1, Traf6 and Sod2 were strongly significant only with this reduce number of samples, we decided not to wait to have new animals to corroborate these experiments, since this would mean a delay in publishing our results.
Reviewer 2 Report
Authors here report on a rare thus important to be investigated human disease called Lafora disease. Their experimental approch, indeed technically sound, was a small RNA-seq and quantitative PCR mediated analysis performed in brain extracts from laforin or malin knock out mice (Epm2a-/- and Epm2b-/- respectively) of 3, 7, 12 and 16 months of age. The new and original finding was a marked overexpression of microRNAs miR-155 and miR-146a in an age-dependent manner. In addition, some reliable correlation was shown between such specific miRNA overexpression and defined genes involved in the regulation of proteostasis, oxidative stress, inflammation and I would add immunity. The organization of the communication paper is good, the results of high interest and the discussion valid and fair.
Some minor suggestions: 1) may the authors comment on the higher overexpression of miR-155 in both KO models? 2) The abbreviations used in the text quite often are not explained, thus making the reading by scientists not expert in the field a bit less easy. 3) It could be useful to add some more details about the various genes whose correlation has been proposed with the specific miRNA overexpression observed.
Author Response
Authors here report on a rare thus important to be investigated human disease called Lafora disease. Their experimental approch, indeed technically sound, was a small RNA-seq and quantitative PCR mediated analysis performed in brain extracts from laforin or malin knock out mice (Epm2a-/- and Epm2b-/- respectively) of 3, 7, 12 and 16 months of age. The new and original finding was a marked overexpression of microRNAs miR-155 and miR-146a in an age-dependent manner. In addition, some reliable correlation was shown between such specific miRNA overexpression and defined genes involved in the regulation of proteostasis, oxidative stress, inflammation and I would add immunity. The organization of the communication paper is good, the results of high interest and the discussion valid and fair.
We thank the reviewer for their high praise and positive reception of our work.
Some minor suggestions: 1) may the authors comment on the higher overexpression of miR-155 in both KO models? 2) The abbreviations used in the text quite often are not explained, thus making the reading by scientists not expert in the field a bit less easy. 3) It could be useful to add some more details about the various genes whose correlation has been proposed with the specific miRNA overexpression observed.
We also thank the reviewer for their insightful comments. Regarding the first question, we think that since miR-155 has been shown to mediate in pro-inflammatory responses, and based in the fact that neuroinflammation is progressively developed in the animal models under study, it is possible that the increase observed might represent one of the first alterations at the microRNA level, and thus the anti-inflammatory processes mediated by miR-146a appear later as a response to the previous increase of miR-155. This could explain the highest levels of miR-155 in both animal models and is in agreement with the overexpression of genes like Socs1 that also respond to pro-inflammatory situations.
To address the second question, we have reviewed the manuscript and defined all abbreviations that could lead to confusion, and hopefully the work is now more understandable by the general public; we thank the reviewer for their advice.
Finally, we have further developed the discussion including some additional information on the genes described (lines 171-182):
“All the genes analyzed are somehow related to molecular pathways relevant for the pathophysiological context of LD: Nsun3 (NOP2/Sun RNA Methyltransferase 3) encodes a mitochondrial methyltransferase related to encephalomyopathy and seizures [18] ; Glra2 (Glycine Receptor Alpha 2) encodes for a glycine receptor that has been predicted to influence cognition, learning and memory in the developing brain [19] ; Btg2 (BTG Anti-Proliferation Factor 2 ) is an anti-proliferative gene that, interestingly, has been postulated as a neuroprotective gene in mouse models of Huntington’s dis-ease[20] ; and Traf6 (TNF receptor-associated factor 6) is an immune response regulator with an E3 ubiquitin ligase activity that has been shown to interact with Sod1 in rat models of amyotrophic lateral sclerosis (ALS) promoting aggregation [21].”
See also lines 190-191: “It is noteworthy the marked overexpression observed for the mitochondrial dismutase Sod2 (Superoxide dismutase 2), and Socs1 (Suppressor of Cytokine Signaling 1),…” and lines 202-204: “Socs1 is a negative regulator of cytokine signalling, induced by the latter, thus pointing to a role in counteracting the effect of miR-155 and the cytokine cascade induced with age in LD animal models (as described in [10]).”
We think that these additions will make the results clearer and thank the reviewer for their suggestions.